# Comparison of perceived family state and functioning among individuals with depression and general population in Southern Thailand

Jarurin Pitanupong[1]*, Kwanpond Traivaranon[1], Napakkawat Buathong[2]

1 Department of Psychiatry, Faculty of Medicine, Prince of Songkla University, Hat Yai, Songkhla, Thailand, 2 Department of Family and Preventive Medicine, Faculty of Medicine, Prince of Songkla University, Hat Yai, Songkhla, Thailand

* pjarurin@medicine.psu.ac.th

## Abstract

### Purpose

This study aimed to examine perceived family functioning among individuals with depression, then compare these perceptions with those of the general population.

### Materials and methods

A cross-sectional study was conducted among individuals with depression at Songklanagarind Hospital and individuals from the general population; from May to July 2024. Participants completed three questionnaires: 1) Personal and demographic inquiry, 2) the Family State and Functioning Assessment Scale (FSFAS-25), and 3) the Patient Health Questionnaire (PHQ-9). Data analysis involved descriptive statistics, Chi-square or Fisher's exact test, Wilcoxon rank sum test and Student's t-test.

### Results

The study compared 41 individuals with depression with 41 from the general population; revealing significant differences in family functioning scores. The depression group reported lower median family functioning scores (76 [IQR 64-84]) compared with the general population (87 [IQR 77-93]). Fewer individuals in the depression group reported high total family functioning scores (56.1% vs. 82.9%, $p = 0.016$). They also showed lower percentages in family support (61.0% vs. 95.1%, $p < 0.001$) and discipline dimensions (46.3% vs. 78.0%, $p = 0.006$). Furthermore, those with residual depression symptoms (PHQ-9 having a score of nine or higher) showed significant differences in family support (37.5% vs. 76.0%, $p = 0.033$) and emotional status (18.8% vs. 60.0%, $p = 0.023$) compared with those without residual depression symptoms.

**Data availability statement:** All relevant data are within the paper and its Supporting information files.

**Funding:** The author(s) received no specific funding for this work.

**Competing interests:** The authors have declared that no competing interests exist.

## Conclusion

Individuals with depression demonstrated lower levels of family functioning compared with the general population. Acknowledging and addressing the influence of family dynamics on the development and persistence of the disorder may be essential for improving treatment outcomes. Integrating these factors into person-centered mental health interventions can lead to more comprehensive, individualized, and effective care.

## Introduction

The prevalence of major depressive disorder (MDD) among Thai individuals, estimated at approximately 1.5 million [1], represents a significant public health concern. Additionally, it impacts the quality of life, physical health and social functioning, while also imposing an economic burden [2,3]. The etiology of MDD is complex, involving biological factors; such as neurotransmitter imbalances and psychosocial influences [4,5]. Although, antidepressants serve as the primary treatment [6], many individuals with MDD still experience residual depression symptoms [7]. This highlights the importance of comprehensive treatment approaches that aim at achieving complete symptom remission [8] and the restoration of pre-illness functioning [9,10]. Recent studies have focused on the role of psychosocial factors in MDD, with childhood maltreatment [11], family dynamics, and social support identified as key contributors to the onset and progression of the disorder [12–15]. Dysfunctional family environments can exacerbate symptoms of MDD and contribute to relapse, influencing family functioning, which is an essential area of investigation [16,17].

Family functioning refers to how family members interact, communicate, and collaborate to resolve issues, meet needs, and support individual well-being [18,19]. Studies have shown that families of individuals with psychiatric disorders often experience dysfunction in areas; such as problem-solving, emotional responsiveness, involvement, communication and behavior control [20]. A study conducted at Songklanagarind Hospital in Southern Thailand found that relatives of individuals with MDD reported low perceived burden and psychological distress, while also experiencing high levels of social support [3]. This indicates that family dynamics can have a positive impact, challenging the view that all familial interactions are detrimental in the context of MDD [21]. While family functioning among individuals with psychiatric disorders tends to be less healthy than the general population [20], many Thai individuals with MDD continue to experience residual depression symptoms [7]. Therefore, family dysfunction; such as poor problem-solving and low emotional responsiveness, may be factors contributing to residual depression symptoms [22].

Most existing studies have focused on Western contexts, which may overlook the unique complexities of Thai families, which are influenced by cultural, social, and religious factors; including Buddhism and Islam. While Buddhism predominantly influences the cultural context in Thailand, the southern region is distinct due to its significant Muslim population.

A Muslim family is often characterized by strong unity and implements love, cooperation, respect and mutual understanding between each other [23]. This multicultural and multi-religious composition may lead to variations in family dynamics compared to both other regions within Thailand as well as in Western countries. Additionally, in Thailand there has been only one study comparing family functioning in families of psychiatric patients to nonclinical families [24]. No previous studies have focused on the families of individuals with MDD. Hence, recognizing these dynamics within the Thai context is essential for developing effective support strategies for Thai families of individuals with MDD. Therefore, this study aimed to: 1) explore the characteristics of family functioning (including family support, discipline, communication, problem-solving, emotional status and relationships), and 2) compare family functioning between individuals with MDD and the general population. Additionally, since some individuals with MDD may experience residual depression symptoms, this study also compared the family functioning characteristics of those with and without these residual depression symptoms. By integrating patient perspectives with existing literature, we can gain a deeper understanding of how family relationships influence MDD. This holistic approach could provide for more effective therapeutic strategies, ultimately enhancing treatment outcomes for individuals with MDD.

## Materials and methods

### Respondents and procedure

After receiving approval from the Ethics Committee of the Faculty of Medicine, at Prince of Songkla University (REC: 66-554-3-4), this cross-sectional study was conducted at a psychiatric outpatient clinic, in Songklanagarind Hospital, an 800-bed tertiary referral center in Southern Thailand; including participants from the Kamphaeng Phet subdistrict Municipality in Rattaphum district. This diverse municipality provides demographic and cultural data that aligns well with individuals in the MDD group, enhancing the study's relevance.

For the MDD group, outpatients visiting the psychiatric clinic from May to July 2024, were invited to participate. Eligible participants were required to have a diagnosis of MDD, be at least 20 years old, be fluent in Thai and be willing to complete questionnaires. Those with other psychiatric diagnoses or comorbidity and the presence of psychotic features were excluded from the study. A thorough evaluation ensured all inclusion and exclusion criteria were met. Participants were informed as to the study's purpose, and given 15–20 minutes to consider their involvement before signing an informed consent form.

For the general population group, volunteers aged 20 and older, also fluent in Thai, were recruited through convenience sampling: they were further screened for any psychiatric history or substance use. Interested individuals received an information sheet, and had 15–20 minutes to decide on participation. Those whom agreed signed consent forms and completed self-administered questionnaires privately; with researchers and health volunteers available for support.

A power analysis was performed to calculate the required sample size to compare the proportion of dysfunctional families between the two groups: individuals with MDD and the general population. The analysis was based on the following parameters: a significance level of 0.05, a power of 0.80, and expected proportions of dysfunctional families of 83.3% in the individuals with MDD group, and 45.0% in the general population group [24]. Assuming equal group sizes ($n_2/n_1 = 1$), the analysis estimated that 29 participants per group (total 58) would be required to detect a significant difference. To ensure data robustness, and account for potential participant drop-out, the sample size was increased to 82 participants (41 per group). Convenience sampling in the Kamphaeng Phet subdistrict Municipality facilitated careful gender and age matching with the MDD group, allowing for a comprehensive comparison of family dynamics and their mental health implications.

### Questionnaire

1. **Personal and demographic information**

Participants provided information on: age, gender, marital status, religion, education level, occupation, income, underlying disease, family type, age onset of depression, type of antidepressants and duration of treatment.

## 2. The Family State and Functioning Assessment Scale (FSFAS-25), Thai version

This is a self-report questionnaire designed to evaluate family state and functioning assessment scale (FSFAS) [25]. It is mostly based on the structural domains of the Family Assessment Device from the McMaster Model of Family Functioning (MMFF) [26], Chulalongkorn family inventory [27], and the Thai family functioning scale [28]. Participants agree with various statements on a 4-point Likert scale; ranging from "1 = strongly disagree" to "4 = strongly agree." The scale features both positive and negative statements, requiring the recording of scores for the negative items before calculating the total score. Positive statements include items 1–16, whereas negative statements consist of items 17–25. The total FSFAS score ranges from 25 to 100 and is obtained by summing the raw scores across all 25 items, with higher scores indicating better family functioning. The scale is organized into five dimensions: family support (items 1–5), discipline (items 6–11), communication and problem-solving (items 12–16), emotional status (items 17–21), and relationships (items 22–25). Each dimension is classified into high, moderate, and low levels.

| | Family functioning levels | | |
|---|---|---|---|
| | Low | Moderate | High |
| Family support | 5-9 | 10-14 | 15 and above |
| Discipline | 6-11 | 12-17 | 18 and above |
| Communication and problem solving | 5-9 | 10-14 | 15 and above |
| Emotional status | 5-9 | 10-14 | 15 and above |
| Relationship | 4-7 | 8-11 | 12 and above |
| Total | 25-49 | 50-74 | 75 and above |

This questionnaire has demonstrated strong internal consistency, with a Cronbach's alpha coefficient of 0.87 for the total, and 0.70–0.84 for the subscales [25]. In this study, the FSFAS-25 exhibited even greater internal consistency, with a Cronbach's alpha coefficient of 0.94 for the total, and 0.72–0.91 for the subscales. This indicated excellent reliability in assessing family dynamics within the Thai context.

## 3. The Patient Health Questionnaire (PHQ-9), Thai version

This is a self-rating tool used to assess depression through nine questions, employing a 4-point scale: never (0), rarely (1), sometimes (2), and always (3). The total score ranges from 0 to 27. Score interpretations are as follows: 0–4 (no/minimal depression), 5–9 (mild depression), 10–14 (moderate depression), 15–19 (moderately severe depression), and 20 or higher (severe depression). The cut-off score of nine or higher indicates the presence of depression. Therefore, in this study, the presence of residual depression symptoms is defined as a PHQ-9 score of nine or higher [7,22]. The PHQ-9 has shown good internal consistency, with a Cronbach's alpha of 0.79, alongside a sensitivity of 0.53 and specificity of 0.98 [29]. In this study, the PHQ-9 demonstrated even greater internal consistency, achieving a Cronbach's alpha coefficient of 0.94, indicating strong reliability for evaluating depression in the Thai context.

## Statistical analysis

Descriptive statistics were calculated using proportions, means, standard deviation (SD), median, and interquartile range (IQR). The data distribution was analyzed using the Shapiro-Wilk test. Variables, with a $p$-value $> 0.05$ being considered as normally distributed; whereas those with a $p$-value $\leq 0.05$ were considered non-normally distributed. Based on these results, appropriate parametric or non-parametric tests were selected for further analysis. The comparison between groups was performed using a Student's t-test, Wilcoxon rank sum test (or Mann Whitney U test), and Chi-square test or Fisher's exact test, depending on the data distribution. All analyses were conducted using R version 4.3.1 (The R Foundation for Statistical Computing, Vienna, Austria). A $p$-value $< 0.05$ was considered to indicate significant statistical differences.

## Results

### Demographic characteristics

From May to July 2024, this study included 82 adults; comprising of those meeting the criteria for MDD (n=41) and a general population group (n=41). The response rates were 97.6% for the MDD group and 100.0% for the general population group. Significant demographic differences between the groups were observed in education, occupation, income and history of physical illness (*p*<0.02) (Table 1).

The MDD group had a mean (SD) PHQ-9 score of 7.8 (6.6), with 61.0% scoring below nine, indicating no residual depression symptoms. The levels of depression are illustrated in Fig 1.

### Comparison of family state and functioning between individuals with depression and general population

The MDD group reported significantly lower median family functioning scores; at 76 (IQR: 64–84), compared with the general population's median score of 87 (IQR: 77–93). When classifying family functioning into high, moderate, and low levels, a notably lower percentage of the MDD group exhibited high total family functioning (56.1%) compared with the general population (82.9%), with a *p*-value of 0.016. Additionally, when examining specific dimensions, the MDD group demonstrated significantly lower percentages in both family support and discipline. Specifically, 61.0% of the MDD group reported high family support, compared with 95.1% in the general population (*p*<0.001). For the discipline dimension, only 46.3% of the MDD group reported high discipline, compared with 78.0% of the general population (*p*=0.006) (Table 2).

Comparing individuals with residual depression symptoms (PHQ-9 score of nine or higher) to those with non- residual depression symptoms revealed significant differences in the family support and emotional status dimensions. Only 37.5% of the residual depression symptoms group reported high family support, compared with 76.0% in the non- residual depression symptoms group (*p*=0.022). Additionally, the emotional status dimension was significantly lower in the residual depression symptoms group, with 18.8% reporting high emotional status, compared to 60.0% in the non- residual depression symptoms group (*p*=0.012) (Table 3).

## Discussion

This study is the first in Thailand to examine perceived family functioning among individuals with MDD and to compare these perceptions with those of the general population. The findings indicated that individuals with MDD reported significantly lower family functioning scores, particularly in dimensions of family support and discipline. Furthermore, those with residual depression symptoms demonstrated even lower levels of family support and emotional status compared with individuals without residual depression symptoms. These results emphasize the importance of family dynamics in MDD as well as highlighting the need for targeted interventions to enhance family functioning within this population.

Our findings align with prior studies, which also indicated a significant association between family dysfunction and MDD [30]. These studies suggest that individuals with psychiatric disorders experience higher rates of family dysfunction compared with those without such disorders [20]. This dysfunction carries a significant psychosocial burden on individuals, emphasizing the important role of family dynamics in the recovery, particularly in dimensions of family support [24].

Family environment variables; such as structure, function, support and conflict, are closely related to disease management, treatment adherence, coping strategies, complications and psychological adaptation for both individuals and families [20]. For instance, a study at Songklanagarind Hospital in Southern Thailand found that individuals with residual depression symptoms reported lower levels of family support than those without [22]. Similarly, our study found that individuals with residual depression symptoms exhibited lower family support and poorer emotional well-being than those without residual depression symptoms. These findings suggest that improving family support may enhance treatment outcomes, reduce psychosocial factors contributing to residual depression symptoms, and enhance overall well-being. This aligns with treatment goals focused on helping individuals with MDD achieve their full recovery without residual depression symptoms [31].

**Table 1. Demographic characteristics (N = 82).**

| Demographic characteristics | Total (N = 82) | Number (%) | | Chi2 *P*-value |
| --- | --- | --- | --- | --- |
| | | **General population (N = 41)** | **Individuals with MDD (N = 41)** | |
| **Gender** | | | | 1 |
| Male | 28 (34.1) | 14 (34.1) | 14 (34.1) | |
| Female | 54 (65.9) | 27 (65.9) | 27 (65.9) | |
| **Age (years)** | | | | 1 |
| 20-30 | 14 (17.1) | 7 (17.1) | 7 (17.1) | |
| 31-40 | 14 (17.1) | 7 (17.1) | 7 (17.1) | |
| 41-50 | 16 (19.5) | 8 (19.5) | 8 (19.5) | |
| 51-60 | 20 (24.4) | 10 (24.4) | 10 (24.4) | |
| >60 | 18 (22.0) | 9 (22.0) | 9 (22.0) | |
| Mean (S.D.) | 46.9 (15.4) | 47.9 (16.0) | 46 (14.9) | 0.584[a] |
| **Status** | | | | 0.507 |
| Single/Divorced | 40 (48.8) | 22 (53.7) | 18 (43.9) | |
| Married | 42 (51.2) | 19 (46.3) | 23 (56.1) | |
| **Religion** | | | | 0.736 |
| Buddhism | 72 (87.8) | 37 (90.2) | 35 (85.4) | |
| Islam | 10 (12.2) | 4 (9.8) | 6 (14.6) | |
| **Education level** | | | | 0.005 |
| Primary education or less | 16 (19.5) | 13 (31.7) | 3 (7.3) | |
| Secondary/High school | 21 (25.6) | 12 (29.3) | 9 (22.0) | |
| Bachelor's degree or higher | 45 (54.9) | 16 (39.0) | 29 (70.7) | |
| **Occupation** | | | | < 0.001 |
| Government officer/ state enterprise employee/ retired government official | 16 (19.5) | 4 (9.8) | 12 (29.3) | |
| Company officer | 5 (6.1) | 1 (2.4) | 4 (9.8) | |
| Employee/ agriculture | 19 (23.2) | 16 (39.0) | 3 (7.3) | |
| Self-employed/ merchant | 23 (28.0) | 17 (41.5) | 6 (14.6) | |
| Student/unemployed | 19 (23.2) | 3 (7.3) | 16 (39.0) | |
| **Income (baht/month)** | | | | <0.001[b] |
| Median (IQR) | 15000 (10000, 25000) | 12000 (10000,15000) | 25000 (17250,50000) | |
| **Physical illness** | | | | 0.009 |
| No | 56 (68.3) | 34 (82.9) | 22 (53.7) | |
| Yes | 26 (31.7) | 7 (17.1) | 19 (46.3) | |
| **Family type** | | | | 0.258 |
| Nuclear family | 50 (61.0) | 22 (53.7) | 28 (68.3) | |
| Extended family | 32 (39.0) | 19 (46.3) | 13 (31.7) | |

**Note**: [a] P-value from t-test; [b] P-value from Wilcoxon rank sum test *(or Mann Whitney U test)*.

S.D. = standard deviation; IQR = interquartile range; there were 21 missing values for income.

However, our study found no significant statistical difference in the communication and problem-solving dimensions between the MDD group and the general population (*p* = 0.087), nor between the residual and non-residual depression symptoms groups (*p* = 0.074). This lack of difference may be attributed to the small sample size limitation. Hence, future studies should involve a larger population to further explore these dimensions.

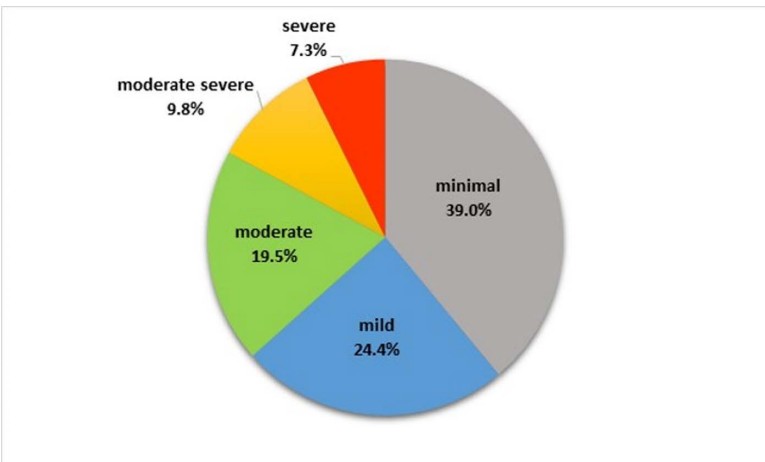

**Fig 1. Level of depression among individuals with depression (N = 41).**

**Table 2. Comparison of family state and functioning between the general population and individuals with depression.**

|  | General population (N = 41) | Individuals with MDD (N = 41) | Chi2 p-value |
|---|---|---|---|
| **Family support** |  |  | < 0.001* |
| Low/ Moderate | 2 (4.9) | 16 (39.0) |  |
| High | 39 (95.1) | 25 (61.0) |  |
| **Discipline** |  |  | 0.006* |
| Low/ Moderate | 9 (22.0) | 22 (53.7) |  |
| High | 32 (78.0) | 19 (46.3) |  |
| **Communication and problem solving** |  |  | 0.087 |
| Low/ Moderate | 4 (9.8) | 11 (26.8) |  |
| High | 37 (90.2) | 30 (73.2) |  |
| **Emotional status** |  |  | 0.121 |
| Low/ Moderate | 15 (36.6) | 23 (56.1) |  |
| High | 26 (63.4) | 18 (43.9) |  |
| **Relationship** |  |  | 0.182 |
| Low/ Moderate | 6 (14.6) | 12 (29.3) |  |
| High | 35 (85.4) | 29 (70.7) |  |
| **Total** |  |  | 0.016* |
| Low/ Moderate | 7 (17.1) | 18 (43.9) |  |
| High | 34 (82.9) | 23 (56.1) |  |

* Significant difference between groups.

A holistic approach is essential for improving outcomes in individuals with MDD. This should include developing effective treatment strategies that integrate family dynamics, ultimately enhancing mental health outcomes. Healthcare professionals should prioritize family assessments and encourage healthy family functioning [32]. For those with residual depression symptoms, therapeutic interventions should extend beyond pharmacotherapy to focus on strengthening family support and emotional well-being; therefore, aiding in the remission of residual depression symptoms. Since both family

**Table 3. Family functioning among individuals with and without residual depression symptoms (N = 41).**

| | PHQ-9 score < 9 (N = 25) | PHQ-9 score ≥ 9 (N = 16) | Fisher's exact test $p$-value |
|---|---|---|---|
| **Family support** | | | 0.022* |
| Low/ Moderate | 6 (24.0) | 10 (62.5) | |
| High | 19 (76.0) | 6 (37.5) | |
| **Discipline** | | | 1 |
| Low/ Moderate | 13 (52.0) | 9 (56.2) | |
| High | 12 (48.0) | 7 (43.8) | |
| **Communication and problem solving** | | | 0.074 |
| Low/ Moderate | 4 (16.0) | 7 (43.8) | |
| High | 21 (84.0) | 9 (56.2) | |
| **Emotional status** | | | 0.012* |
| Low/ Moderate | 10 (40.0) | 13 (81.2) | |
| High | 15 (60.0) | 3 (18.8) | |
| **Relationship** | | | 0.485 |
| Low/ Moderate | 6 (24.0) | 6 (37.5) | |
| High | 19 (76.0) | 10 (62.5) | |
| **Total** | | | 0.106 |
| Low/ Moderate | 8 (32.0) | 10 (62.5) | |
| High | 17 (68.0) | 6 (37.5) | |

**Note:**

*Significant difference between groups.

dysfunction and MDD are multifactorial conditions, comprehensive assessments are essential to identify the various contributing factors [30,31].

To our knowledge, this is the only study conducted in Southern Thailand on this topic within the past decade. The study focused on comparing family functioning between families of individuals with MDD and those from the general population. Additionally, this study explored differences in family functioning between families of individuals with residual depression symptoms and those without: a comparison that has not been previously examined.

However, the study has some limitations; including its quantitative approach and reliance on self-administered questionnaires to assess how individuals with MDD perceive their family environments. This approach may introduce biases, as some participants might report negative perceptions of family support influenced by their symptoms of depression. Notably, 7.3% of the MDD group reported severe MDD. While the sample size was sufficient for statistical power, the general population group was obtained through convenience sampling from the Kamphaeng Phet subdistrict Municipality, which may not fully represent the broader Southern Thai population. Cultural and regional differences could influence family dynamics, so generalizing the findings to other regions or countries may not be entirely appropriate without further study. In addition, significant demographic differences in education levels, occupations, incomes, and histories of physical illness were observed between the MDD and general population groups. As a result, physical illness, education level and socioeconomic factors should be considered as potential confounders.

Furthermore, the primary objective of our study was to examine perceived family functioning among individuals with MDD and to compare it with that of the general population, rather than identify associated factors. Consequently, our analysis focused on direct group comparisons without adjusting for confounders. However, the absence of adjustments may influence the magnitude and direction of the observed differences, potentially leading to over- or underestimation of family functioning differences between the groups.

Finally, future studies should focus on a longitudinal design, involve a larger and more diverse population, and use a representative sample, ideally employing a multi-center approach and including participants from various regions of Thailand. Additionally, designing a prospective therapeutic study that includes family interventions targeting specific aspects of family functioning; such as family support, discipline, emotional status, and long-term follow-ups, could provide valuable insights for treatment planning. Applying more advanced statistical techniques; including multivariate approaches, to explore the role of potential confounders in greater depth could enhance the understanding of family dynamics in MDD.

## Conclusion

Individuals with MDD demonstrate lower family functioning compared with the general population. Recognizing and addressing the role of family functioning is essential in developing comprehensive, person-centered mental health interventions for individuals with MDD. However, this study is limited by confounding variables, the inability to determine the directionality of associations (i.e., reverse causality), and issues with generalizability. Therefore, future research should focus on specific dimensions of family functioning accompanied by long-term follow-up assessments. These interventions could provide critical insights into effective treatment planning by highlighting how family dynamics influence recovery.

## Acknowledgments

The authors would like to acknowledge the participants for their willingness to provide information. We would like to also acknowledge the team's research assistants; Professor Hutcha Sriplung, Nisan Werachattawan and Kruewan Jongborwanwiwat for their support. The English of this article was proofread/edited by the Office of International Affairs, Faculty of Medicine, Prince of Songkla University.

## Author contributions

**Conceptualization:** Jarurin Pitanupong, Kwanpond Traivaranon, Napakkawat Buathong.

**Data curation:** Jarurin Pitanupong, Kwanpond Traivaranon, Napakkawat Buathong.

**Formal analysis:** Jarurin Pitanupong, Kwanpond Traivaranon, Napakkawat Buathong.

**Investigation:** Jarurin Pitanupong, Kwanpond Traivaranon, Napakkawat Buathong.

**Methodology:** Jarurin Pitanupong, Kwanpond Traivaranon, Napakkawat Buathong.

**Project administration:** Kwanpond Traivaranon.

**Resources:** Kwanpond Traivaranon.

**Supervision:** Jarurin Pitanupong.

**Validation:** Jarurin Pitanupong, Kwanpond Traivaranon, Napakkawat Buathong.

**Visualization:** Jarurin Pitanupong, Kwanpond Traivaranon.

**Writing – original draft:** Jarurin Pitanupong, Kwanpond Traivaranon, Napakkawat Buathong.

**Writing – review & editing:** Jarurin Pitanupong.

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
