## [Decision Letter · Decision Letter 0]

6 Feb 2025

PONE-D-24-52399Comparison of perceived family state and functioning among individuals with depression and general populationin Southern ThailandPLOS ONE

Dear Dr. Pitanupong,

Thank you for submitting your manuscript to PLOS ONE. After careful consideration, we feel that it has merit but does not fully meet PLOS ONE’s publication criteria as it currently stands. Therefore, we invite you to submit a revised version of the manuscript that addresses the points raised during the review process.

I agree with the reviewers about the need for revisions of the manuscript.

We look forward to receiving your revised manuscript.

Kind regards,

Diego A. Forero, MD; PhD

Academic Editor

PLOS ONE

Journal requirements:   When submitting your revision, we need you to address these additional requirements. 1. Please ensure that your manuscript meets PLOS ONE's style requirements, including those for file naming. The PLOS ONE style templates can be found at https://journals.plos.org/plosone/s/file?id=wjVg/PLOSOne_formatting_sample_main_body.pdf and https://journals.plos.org/plosone/s/file?id=ba62/PLOSOne_formatting_sample_title_authors_affiliations.pdf. 2. Please match your authorship list in your manuscript file and in the system. 3. In the online submission form, you indicated that [This manuscript does not report data generation or analysis. However, it is available on request from the corresponding author via e-mail: pjarurin@medicine.psu.ac.th.]. All PLOS journals now require all data underlying the findings described in their manuscript to be freely available to other researchers, either 1. In a public repository, 2. Within the manuscript itself, or 3. Uploaded as supplementary information.This policy applies to all data except where public deposition would breach compliance with the protocol approved by your research ethics board. If your data cannot be made publicly available for ethical or legal reasons (e.g., public availability would compromise patient privacy), please explain your reasons on resubmission and your exemption request will be escalated for approval.  4. We note that you have indicated that there are restrictions to data sharing for this study. For studies involving human research participant data or other sensitive data, we encourage authors to share de-identified or anonymized data. However, when data cannot be publicly shared for ethical reasons, we allow authors to make their data sets available upon request. For information on unacceptable data access restrictions, please see http://journals.plos.org/plosone/s/data-availability#loc-unacceptable-data-access-restrictions.  Before we proceed with your manuscript, please address the following prompts: a) If there are ethical or legal restrictions on sharing a de-identified data set, please explain them in detail (e.g., data contain potentially identifying or sensitive patient information, data are owned by a third-party organization, etc.) and who has imposed them (e.g., a Research Ethics Committee or Institutional Review Board, etc.). Please also provide contact information for a data access committee, ethics committee, or other institutional body to which data requests may be sent. b) If there are no restrictions, please upload the minimal anonymized data set necessary to replicate your study findings to a stable, public repository and provide us with the relevant URLs, DOIs, or accession numbers. Please see http://www.bmj.com/content/340/bmj.c181.long for guidelines on how to de-identify and prepare clinical data for publication. For a list of recommended repositories, please see https://journals.plos.org/plosone/s/recommended-repositories. You also have the option of uploading the data as Supporting Information files, but we would recommend depositing data directly to a data repository if possible. Please update your Data Availability statement in the submission form accordingly. 5. Your ethics statement should only appear in the Methods section of your manuscript. If your ethics statement is written in any section besides the Methods, please delete it from any other section.  6. Please remove your figures from within your manuscript file, leaving only the individual TIFF/EPS image files, uploaded separately. These will be automatically included in the reviewers’ PDF.

Reviewers' comments:

Reviewer's Responses to Questions

**Comments to the Author**

1. Is the manuscript technically sound, and do the data support the conclusions?

Reviewer #1: Yes

Reviewer #2: Partly

2. Has the statistical analysis been performed appropriately and rigorously? 

Reviewer #1: Yes

Reviewer #2: No

3. Have the authors made all data underlying the findings in their manuscript fully available?

Reviewer #1: Yes

Reviewer #2: No

4. Is the manuscript presented in an intelligible fashion and written in standard English?

Reviewer #1: Yes

Reviewer #2: Yes

5. Review Comments to the Author

Reviewer #1: The manuscript entitled “Comparison of Perceived Family State and Functioning Among Individuals with Depression and the General Population in Southern Thailand” addresses a highly relevant topic in the field of mental health: the relationship between family functioning and Major Depressive Disorder (MDD). The study establishes a clear and well-defined objective from the outset: to compare family functioning between individuals with MDD and the general population, as well as to explore differences between individuals with residual and non-residual MDD. This clarity enables the manuscript to be structured in an orderly and coherent manner, ensuring that the results directly address the stated objectives.

The study adopts a sound methodology, employing a cross-sectional design with validated instruments such as the PHQ-9 and FSFAS-25, which ensure the reliability of the data collected. The statistical analysis is appropriately conducted using Chi-square, Fisher’s exact, and Student’s t-tests, allowing for the identification of significant differences between groups and strengthening the validity of the findings. Moreover, the study adheres strictly to international ethical standards, having received approval from an ethics committee and complying with the Declaration of Helsinki, which reinforces the scientific and ethical integrity of the work.

Reviewer #2: Major Revisions

Introduction and clarity of hypotheses

- The introduction does not clearly delineate how this study advances knowledge. A more explicit discussion of gaps in prior Thai and international research is needed.

- The hypotheses are not clearly stated as testable assertions. The manuscript should include explicit statements.

- The study does not specify which dimensions of family functioning (e.g., support, discipline) are expected to differ significantly. A dedicated section on objectives and hypotheses should be added for clarity.

- The manuscript does not clearly state the sources for FSFAS-25 and PHQ-9. Are there validated Thai versions of these scales available?

Statistical Analysis

- The study does not employ multivariate techniques (e.g., regression, ANCOVA) to adjust for demographic variables such as income and education, which may confound the relationship between depression and family functioning. Without a multivariate model (e.g., multiple regression or ANCOVA), it is impossible to determine whether the observed differences in family functioning scores are truly due to depression or are confounded by socioeconomic factors.

- The arbitrary cutoff (≥9 vs. <9) oversimplifies depressive symptoms, and the small residual depression subgroup (n=16) reduces statistical power, increasing the risk of Type II errors. The threshold of 9, while clinically valid, may oversimplify the continuum of depressive symptoms. A more nuanced analysis (e.g., treating PHQ-9 as a continuous variable) would provide better insight.

- Tests such as the t-test and Chi-square assume normality of data distribution and sufficient sample sizes per category. For small groups (e.g., n=16 in the residual depression subgroup), these assumptions may be violated, leading to unreliable p-values. Non-parametric alternatives (e.g., Mann-Whitney U test) are more appropriate but underpowered for small samples. While the Shapiro-Wilk test for normality is mentioned, the results are not reported in the manuscript, leaving uncertainty about whether appropriate tests were chosen for the data.

- The manuscript emphasizes significant findings (e.g., family support differences) without adequately addressing the effect sizes. Reporting effect sizes alongside p-values would help readers understand the practical significance of the observed differences.

Discussion Section

- Statements about family functioning’s role in depression recovery are too broad given methodological limitations. Conclusions should be framed more cautiously. The small sample size (41 individuals per group) obtained through convenience sampling significantly limits the generalizability of findings. Furthermore, the division of the sample into subgroups (PHQ-9 ≥9 vs. <9) compounds the problem, as the statistical power decreases substantially when analyzing smaller subgroups. These limitations should have been explicitly addressed in the discussion and considered in the design stage.

- The discussion relies heavily on Thai studies, limiting global relevance. A broader literature review is necessary.

- The possibility that depressive symptoms distort self-perceptions of family functioning should be addressed. Consideration of physical illness and socioeconomic confounders is also necessary.

- Recommendations for family-focused interventions are vague. The manuscript should specify evidence-based approaches that could be implemented in practice.

Minor Revisions

Language and Clarity

- The manuscript contains errors and awkward phrasing. Examples:

"compare these perceptions to those of the general population" → should be "compare these perceptions with those of the general population."

"highlighting the importance of comprehensive treatment approaches aimed at achieving complete symptom remission" → should be revised for conciseness.

- Improve clarity and consistency in reporting statistical findings.

References

- Several references (e.g., Lines 304–312) should be updated with recent studies on family dynamics and depression.

- Ensure all citations follow the required journal style.

6. PLOS authors have the option to publish the peer review history of their article (what does this mean? ). If published, this will include your full peer review and any attached files.

**Do you want your identity to be public for this peer review?** For information about this choice, including consent withdrawal, please see our Privacy Policy .

Reviewer #1: No

Reviewer #2: **Yes: ** Martina Camelio

---

## [Author Response · Author response to Decision Letter 0]

13 Feb 2025

We would like to thank the editor and reviewers for both of your kind advice and help in making this manuscript more academically valuable. And we hope to receive kindness from the honorable recommendation again.

---

## [Decision Letter · Decision Letter 1]

16 Mar 2025

PONE-D-24-52399R1Comparison of perceived family state and functioning among individuals with depression and general populationin Southern ThailandPLOS ONE

Dear Dr. Pitanupong,

Thank you for submitting your manuscript to PLOS ONE. After careful consideration, we feel that it has merit but does not fully meet PLOS ONE’s publication criteria as it currently stands. Therefore, we invite you to submit a revised version of the manuscript that addresses the points raised during the review process.

I agree with both reviewers about the need for further revisions.

We look forward to receiving your revised manuscript.

Kind regards,

Diego A. Forero, MD; PhD

Academic Editor

PLOS ONE

Journal Requirements:

Reviewers' comments:

Reviewer's Responses to Questions

**Comments to the Author**

1. If the authors have adequately addressed your comments raised in a previous round of review and you feel that this manuscript is now acceptable for publication, you may indicate that here to bypass the “Comments to the Author” section, enter your conflict of interest statement in the “Confidential to Editor” section, and submit your "Accept" recommendation.

Reviewer #1: (No Response)

Reviewer #3: (No Response)

2. Is the manuscript technically sound, and do the data support the conclusions?

Reviewer #1: Partly

Reviewer #3: Partly

3. Has the statistical analysis been performed appropriately and rigorously? 

Reviewer #1: (No Response)

Reviewer #3: I Don't Know

4. Have the authors made all data underlying the findings in their manuscript fully available?

Reviewer #1: Yes

Reviewer #3: Yes

5. Is the manuscript presented in an intelligible fashion and written in standard English?

Reviewer #1: Yes

Reviewer #3: Yes

6. Review Comments to the Author

Reviewer #1: The authors have provided a more detailed discussion of gaps in prior research, particularly within the Thai context, emphasizing the influence of cultural and religious factors on family dynamics. While they have formulated clearer and more testable hypotheses, the specific dimensions of family functioning expected to differ (e.g., support, discipline) are not explicitly outlined.

At the methodological level, the authors' approach has several strengths but also notable limitations that may impact the validity and generalizability of the results. A key concern is the absence of effect sizes, which limits readers' ability to assess the practical significance of the findings. Although the sample size was calculated to ensure statistical power, it remains relatively small, particularly when divided into subgroups (e.g., those with PHQ-9 ≥9 vs. <9). This reduction in sample size diminishes statistical power and increases the risk of type II errors (i.e., failing to detect true differences). For instance, the subgroup with residual symptoms (PHQ-9 ≥9) comprises only 16 participants, which restricts the generalizability of the findings. To enhance the study's robustness, future research could consider a longitudinal design, a larger and more diverse sample, and the application of more advanced statistical techniques.

The authors have appropriately moderated their conclusions and acknowledged the study's limitations, such as the small sample size and reliance on convenience sampling. They have also expanded the discussion to include international studies, though the literature review remains predominantly focused on Thai research. Additionally, they have addressed the potential for depressive symptoms to distort self-perceptions of family functioning, which strengthens the manuscript's critical perspective.

Reviewer #3: Thank you for taking the time to revise the manuscript and respond to the reviewers questions.

Of note, for some comments the answers provided were not yet sufficient to respond to the reviewers' requests. I will specify the concerns below:

Answer to Reviewer 1:

For the comment regarding "the hypothesis are not clearly stated": . The aims of the study were mentioned, yet, they seems incomplete as they failed to mention the aim that patients with MDD with residual symptoms would be compared to those without residual symptoms.

The manuscript now describes the dimensions of the construct family functioning, yet for those not familiar with the scale, we cannot tell if each of the 5 domains have the same weight in the overall score (do all 5 domains contribute with the same number of questions?). Also, regarding this variable, only information on the scale range is provided, but for results in table 2 and 3, the results for family functioning (overall and its different components) were dichotomized in low or moderate versus high family support (for example). Which cut-off was used to create these groups and why? Please add that info in the method section.

Regarding the statistical analyses, reviewer question why authors did not use multivariate techniques to adjust for confounders. I believe that the acknowledgement that those factors were confounders in the discussion is not sufficient to address the reviewer's comment. Since you have information on the variables that were potential confounders, it is unclear why you did not assess the role of those factors in the comparisons performed. Also, there is not discussion on what the is the potential impact on the magnitude and directionality of the results because of the "lack of adjustment" for those factors. If there is a reasons justifying why that adjustment was not done, it was not presented in the discussion. Thus, the conclusions and recommendations needs to adjusted accordingly.

On a extra note: Method section would benefit from having further details on sample size calculation (power and alpha error estimated and expected "effect size" should be mentioned. Also, regarding sample selection, further details on how many cases of MDD / community controls were contacted and how many accepted to participate (the response "rate") should be provided so that readers can assess potential for selection biases risks.

Discussion section: The point proposed by the reviewer on possibility of depressive symptoms distorting self-perception of family function in those with MDD is likely the biggest limitation of the present manuscript and needs to be discussed in greater depth. The present study does not allow teasing apart temporality of the association between MDD and lower family functioning (whether depression was the cause of the patient reporting lower family functioning or if lower family functioning lead to MDD or MDD with worse control). Consider discussing findings from prospective studies to provide an idea of the potential impact of such limitation on the findings. If recall bias and/or reverse causality biases are the main reason for the findings hereby reported, then the implications are that this study findings should be taken in light of those limitations and possibly as major implications they should propose future prospective studies. The point is that the implications now are too strong as if there were no major issues in the interpretation of the findings hereby presented.

Reviewer proposed that the recommendations regarding interventions should be evidence-based. Authors did not addressed that completely. Interventions to be assessed were mentioned, but no citations to back up the efficacy/effectiveness of such interventions were provided.

Answer to reviewer 2:

Language and clarity: Partially solved. I suggest further editorial review to improve text. Some terms used does not seem accurate.

For example lines:

110 "They underwent training on study procedures" - this is the type of statement that applies best to data collection, or researchers, but not to participants who take surveys.

115 " the sample calculation used in the literature review". It does not make sense as sample size is not "used" for literature review purposes.

line 119 and 120 - reliability and accuracy terms used as synonyms

line 77: "residual depression". Consider keeping the wording residual depression symptoms", which would be most appropriate.

7. PLOS authors have the option to publish the peer review history of their article (what does this mean? ). If published, this will include your full peer review and any attached files.

**Do you want your identity to be public for this peer review?** For information about this choice, including consent withdrawal, please see our Privacy Policy .

Reviewer #1: No

Reviewer #3: No

---

## [Author Response · Author response to Decision Letter 1]

22 Mar 2025

Thank you so much for your valuable suggestions, which have made our manuscript more scientifically rigorous.

---

## [Decision Letter · Decision Letter 2]

12 Apr 2025

PONE-D-24-52399R2Comparison of perceived family state and functioning among individuals with depression and general population in Southern ThailandPLOS ONE

Dear Dr. Pitanupong,

Thank you for submitting your manuscript to PLOS ONE. After careful consideration, we feel that it has merit but does not fully meet PLOS ONE’s publication criteria as it currently stands. Therefore, we invite you to submit a revised version of the manuscript that addresses the points raised during the review process.

 I agree with the reviewer about the need for an additional minor revision.

We look forward to receiving your revised manuscript.

Kind regards,

Diego A. Forero, MD; PhD

Academic Editor

PLOS ONE

Journal Requirements:

Reviewers' comments:

Reviewer's Responses to Questions

**Comments to the Author**

1. If the authors have adequately addressed your comments raised in a previous round of review and you feel that this manuscript is now acceptable for publication, you may indicate that here to bypass the “Comments to the Author” section, enter your conflict of interest statement in the “Confidential to Editor” section, and submit your "Accept" recommendation.

Reviewer #3: (No Response)

2. Is the manuscript technically sound, and do the data support the conclusions?

Reviewer #3: Partly

3. Has the statistical analysis been performed appropriately and rigorously? 

Reviewer #3: Yes

4. Have the authors made all data underlying the findings in their manuscript fully available?

Reviewer #3: Yes

5. Is the manuscript presented in an intelligible fashion and written in standard English?

Reviewer #3: Yes

6. Review Comments to the Author

Reviewer #3: Thank you for considering the suggestions.

There are two issues that I believe deserve final adjustments.

First, there were some inconsistencies in the text regarding the FSAS score.

In line 148 it reads " FSFAS score, which ranges from 30 to 120... is computed by summing the raw scores across all 30 items,". Yet, the table added follows with inconsistent scores (ranging from the lowest value of 25). Is the range from 25 to 120, then? Also, the 5 dimensions explanation in line 150-152 reads "The scale is organized into five dimensions: family support (items 1-5), discipline (items 6-11) communication and problem-solving (items 12-16), emotional status (items 17-21), and relationships (items 22-25). Where are items 26-30? Which dimension do they belong to?

Finally, as mentioned previously, the results might be the result of methodological limitations (ex: lack of adjustment for confounders, lack of ability to assess directionality of association (reverse-causality), generalizability. Yet, the recommendation in lines 302-303 "To improve outcomes, reduce the likelihood of residual depression symptoms,

and prevent relapse, healthcare providers should implement strategies that enhance family functioning as part of their holistic care." seems too strong. You should consider that your findings might not hold once limitations are solved with other studies. Thus, as proposed before, consider "toning down, for instance, by limiting your recommendation to emphasize the need for further longitudinal studies that would account for the limitations discussed.

7. PLOS authors have the option to publish the peer review history of their article (what does this mean? ). If published, this will include your full peer review and any attached files.

**Do you want your identity to be public for this peer review?** For information about this choice, including consent withdrawal, please see our Privacy Policy .

Reviewer #3: No

---

## [Editor Report · Decision Letter 3]

16 Apr 2025

Comparison of perceived family state and functioning among individuals with depression and general population in Southern Thailand

PONE-D-24-52399R3

Dear Dr. Pitanupong,

We’re pleased to inform you that your manuscript has been judged scientifically suitable for publication and will be formally accepted for publication once it meets all outstanding technical requirements.

Kind regards,

Diego A. Forero, MD; PhD

Academic Editor

PLOS ONE
---

## [Editor Report · Acceptance letter]

PONE-D-24-52399R3

PLOS ONE

Dear Dr. Pitanupong,

I'm pleased to inform you that your manuscript has been deemed suitable for publication in PLOS ONE. Congratulations! Your manuscript is now being handed over to our production team.

Kind regards,

on behalf of

Dr. Diego A. Forero

Academic Editor

PLOS ONE